# Shiga Toxin-Associated Hemolytic Uremic Syndrome: A Narrative Review

**DOI:** 10.3390/toxins12020067

**Published:** 2020-01-21

**Authors:** Adrien Joseph, Aurélie Cointe, Patricia Mariani Kurkdjian, Cédric Rafat, Alexandre Hertig

**Affiliations:** 1Department of Nephrology, AP-HP, Hôpital Tenon, F-75020 Paris, France; adrien.joseph@hotmail.fr (A.J.); cedric.rafat@aphp.fr (C.R.); 2Department of Microbiology, AP-HP, Hôpital Robert Debré, F-75019 Paris, France; aurelie.cointe@aphp.fr (A.C.); patricia.mariani@aphp.fr (P.M.K.); 3Department of Renal Transplantation, Sorbonne Université, AP-HP, Hôpital Pitié Salpêtrière, F-75013 Paris, France

**Keywords:** Shiga toxin, *Escherichia coli*, hemolytic uremic syndrome, thrombotic microangiopathy

## Abstract

The severity of human infection by one of the many Shiga toxin-producing *Escherichia coli* (STEC) is determined by a number of factors: the bacterial genome, the capacity of human societies to prevent foodborne epidemics, the medical condition of infected patients (in particular their hydration status, often compromised by severe diarrhea), and by our capacity to devise new therapeutic approaches, most specifically to combat the bacterial virulence factors, as opposed to our current strategies that essentially aim to palliate organ deficiencies. The last major outbreak in 2011 in Germany, which killed more than 50 people in Europe, was evidence that an effective treatment was still lacking. Herein, we review the current knowledge of STEC virulence, how societies organize the prevention of human disease, and how physicians treat (and, hopefully, will treat) its potentially fatal complications. In particular, we focus on STEC-induced hemolytic and uremic syndrome (HUS), where the intrusion of toxins inside endothelial cells results in massive cell death, activation of the coagulation within capillaries, and eventually organ failure.

## 1. Introduction

Shiga toxin-producing *Escherichia coli*-associated hemolytic uremic syndrome (STEC-HUS) belongs to the body of thrombotic microangiopathies [1], a heterogeneous group of diseases characterized by a triad of features: thrombocytopenia, mechanical hemolytic anemia with schistocytosis, and ischemic organ damage. It is caused by gastrointestinal infection by a Shiga toxin-producing *E. coli* (and occasionally other pathogens) and is also called “typical” HUS, as opposed to “atypical” HUS, which results from alternative complement pathway dysregulation, and “secondary” HUS, caused by various co-existing conditions (see [2,3] and Figure 1).

### 1.1. Historical Perspective

Swiss pediatric hematologist Conrad von Gasser introduced in a paper published in 1955 the term “hemolytic uremic syndrome” [5], but it was not until 1983 that Karmali and colleagues linked the sporadic post-diarrheal HUS of hitherto unknown origin to a toxin produced by specific strains of *E. coli* that they found in the stools of affected children. This toxin was toxic to Vero cells (a line of renal epithelial cells isolated from the African green monkey), hence the name Verotoxin [6]. The same year, Dr. O’Brien and colleagues purified a lethal toxin from the *E. coli* O157:H7 strain, which structurally resembled that of *Shigella dysenteriae* type 1, and termed it Shiga toxin [7]. Both terms still apply to describe the disease, which accounts for an estimated 2,801,000 acute illnesses annually and leads to 3890 cases of HUS [8]. The unprecedented German outbreak of 2011, which led to 3816 cases, including 845 HUS and 54 deaths caused by the emergence of hypervirulent O104:H4, recently acted as a grim reminder of the potentially devastating consequences of STEC-HUS [9].

### 1.2. Purpose of the Review

In this review, we summarize epidemiology, pathophysiology, diagnostic, and treatment measures of STEC-HUS. We emphasize key messages derived from recent outbreaks and advances in the understanding of the pathogenesis that have uncovered potential avenues for future therapies. Other Shiga toxin-producing bacteria (*S. dysenteriae* [10], *S. flexneri* [11,12], *S. sonnei* [13], and *Citrobacter freundii* [14]) and neuraminidase-producing bacteria [15,16] (*Clostridium perfringens* and *Streptococcus pneumoniae*), responsible for rare cases of enteropathic and non-enteropathic infection-induced HUS, are described elsewhere and are beyond the scope of this review.

## 2. Epidemiology and Microbiology

Since the first recorded outbreaks in 1983 [17,18], significant efforts have been made to understand the epidemiology, microbiology and mode of transmission of Shiga toxin-producing *E. coli*. Such efforts concretized in the creation of surveillance networks such as Foodnet in North America, the European Center for Disease Prevention and Control (ECDC), or PulseNet, a global network dedicated to laboratory-based surveillance for food-borne diseases in 85 countries [19].

### 2.1. The Infectious Agent

#### 2.1.1. Nomenclature: Shiga Toxin, Vero Toxin-Producing, or Enterohemorrhagic *E. coli*

The term Shiga toxin-producing *Escherichia coli* (STEC) refers to an *E. coli* strain that acquired the capacity to produce a Shiga toxin, through transfer of gene by means of a Shiga-toxin (Stx) phage. However, not all STEC can infect humans, and only a subset of these are responsible for human disease and belong to the pathovar called enterohemorrhagic *E. coli* (EHEC) [20]. Shiga toxins are also commonly referred to as Verotoxins, a synonym which will not be used in this review. Most EHEC harbor a chromosomal pathogenicity island called *locus of enterocyte effacement* (LEE), encoding, in particular, a type III secretion system (T3SS), an adhesin called intimin, and its receptor Tir. Intimin encoded by the *eae* gene allows for intimate attachment of the bacteria to the intestinal epithelium causing characteristic attaching and effacing lesions and shared with enteropathogenic *E. coli* (EPEC) strains. Enterohemorrhagic *E. coli* harboring LEE are referred to as typical EHEC and those which do not as atypical EHEC. Atypical EHEC possess other adhesion factors such as the STEC autoagglutinating adhesin (Saa) or the AggR transcriptional regulator, which is characteristic of enteroaggregative *E. coli* (EAEC) and were present in the epidemic O104:H4 EHEC involved in the German outbreak [21]. The presence of the intimin (*eae*) gene is associated with human disease and evolution towards hemorrhagic colitis and HUS [22,23]. Several classifications of Shiga toxin-producing *E. coli* have been proposed. Karmali et al. divided STEC into five seropathotypes (A through E) according to their pathogenicity in humans [24], whereas Kobayashi et al. individualized eight clusters based on virulence gene profiles [25]. Nomenclature of *E. coli* and thrombotic microangiopathies is schematized in Figure 1.

#### 2.1.2. Evolution of *E. coli* and Phage Acquisition of Stx Gene

Enterohemorrhagic *E. coli* constitutes a homogeneous pathotype but consists of various phylogenies that have acquired virulence factors (VFs) independently [26]. For example, *E. coli* O157:H7 is believed to have evolved in a series of steps from O55:H7, a recent ancestor of the enteropathogenic serotype associated with infantile diarrhea [27,28]. Unlike *S. dysenteriae* type 1, the capacity of STEC to produce Shiga toxins results from the integration of the genome encoded in various bacteriophages related to phage lambda, called Stx phages [29], in a process known as transduction. These bacteriophages can be cryptic during their lysogenic phase, duplicating with every subsequent cell division of its host, or active and propagate from one receptive enterobacteria to another during their lytic phase [30]. A single STEC strain may carry up to six Shiga toxin-encoding genes [30,31,32]. Shiga toxin is under the control of the phage’s late genetic circuitry and upstream of the lysis cassette. During the lysogenic phase, the expression of most phage genes is inhibited. Certain triggers, in particular SOS-inducing agents such as some antibiotics [33], have the potential to derepress the transcription of phage genes, including Stx, hence switching cells from a lysogenic to lytic phase (induction) [34]. Stx is then released in the extracellular milieu when phage-mediated bacterial lysis occurs [35].

#### 2.1.3. EHEC: Microbiological Characteristics of Classic O157:H7 and Emergent Non-O157 Serotypes

Hundreds of STEC serotypes have been described based on their somatic (O) and flagellar (H) antigens, dozens of which are implicated in human diseases [25]. The first ever to be described was O157:H7 and it remains the predominant serotype to this day, responsible for more than one million cases of diarrhea and an estimated 2000 cases of STEC-HUS worldwide annually [8]. Classic O157:H7 *E. coli* lost its capacity to ferment sorbitol [36], contrary to most commensal and other pathogenic *E. coli*. However, the existence of a sorbitol-fermenting, nonmotile strain has been identified and incriminated in several outbreaks in central Europe. This strain displays enhanced virulence with a greater risk of HUS (30%), requirement for dialysis, and higher case-fatality (11%) [37,38,39,40]. More recently, there is a growing awareness that non-O157 serogroups can also cause severe diseases thanks to the increased availability of immunoassay and molecular tools that allow their detection [41,42]. In northern America and Europe, non-O157 serogroups are increasingly associated with post-STEC-HUS and since the 2010s have even exceeded the number of O157:H7 infections [4,41,43]. Moreover, an unusual serogroup, O80, is currently emerging in France [44,45,46] and Europe [47,48,49]. Several epidemiological and clinical features differentiate O157 and non-O157 serogroups. As a whole, non-O157 serogroups are less associated with outbreaks, are more strongly connected to international travel, and appear to be less prone to elicit STEC-HUS (1% versus 11% risk of STEC-HUS, *p* < 0.001) [41,43,50]. Yet, non-O157 serogroups represent a heterogeneous group, the O104:H4 epidemic serving as an example that these serotypes may also have dreadful consequences [9]. The O104:H4 serotype stands out as one of the most virulent strains responsible for HUS in history (Figure 2). During the German outbreak in 2011, 855 patients suffered from O104:H4-associated STEC-HUS, and over 100 were admitted to intensive care units [51] with more than 50 fatalities recorded [9]. From a microbiological point of view, the O104:H4 serotype also displays unique features. First, despite glaring evidence of its noxious clinical impact, it lacked the canonical VFs encoded in the locus of enterocyte effacement of other EHEC (see Section 3.1). Second, genome-wide comparisons suggested that this strain derived from an enteroaggregative *E. coli,* which acquired a prophage-encoding Shiga toxin 2 and a distinct set of additional virulence and antibiotic-resistance factors via horizontal genetic exchange [52]. It thus combined pathogenic features from enteroaggregative *E. coli*, the capacity to produce Shiga toxin, and an extended-spectrum β-lactamase phenotype. The lack of previous immunity may have acted as an additional factor in the severity of this outbreak. The O26:H11 serotype has emerged as the most common non-O157 serotype causing human disease in Europe [4,53] and North America [42,43]. More specifically, a strain harboring Stx_2_ accounted for approximately 50% of all Stx2a-harboring EHEC O26 strains isolated between 1996 and 2012 in Europe [54]. This serotype has been most commonly found among young children [53] and does not differ from the O157 serotype in terms of severity of disease.

### 2.2. Shiga Toxins: Structure and Nomenclature

Shiga toxins are named after the Japanese microbiologist Kiyoshi Shiga who in 1898 described the bacteria *S. dysenteriae* [55,56]. This bacterium produces a toxin structurally and antigenically identical to *E. coli*-produced Stx1. Shiga toxins are an AB_5_ toxin type consisting of a monomeric, enzymatically active A subunit non-covalently linked to a pentameric B subunit responsible for binding to the glycosphingolipid globotriaosylceramide (Gb3, also known as CD77 or Pk blood group antigen), a specific receptor on the cell surface [57]. Functionally, the Shiga toxins belong to the family of ribosome-inactivating proteins [58]. This AB class of bacterial toxins also includes the pertussis and diphtheria toxins, as well as the cholera toxin family, with which Shiga toxins seem to share a distant evolutionary relationship [59,60]. In addition to their ribosome-modifying properties, Shiga toxins exert various other cellular effects, detailed in Section 3.2. Shiga toxins exist as two immunologically distinct types, Stx1 and Stx2, that share the same structure and function but are not cross-neutralized with heterologous antibodies because of their only 50% homology and 10 subtypes (Stx1a, Stx1c, Stx1d, and Stx2a to Stx2g). Each subtype is then divided into variants that differ from the prototype by one or more amino acids [61]. This nomenclature reflects both the phylogeny and origin of the toxin as well as its pathogenicity. For example, the presence of Stx2 is strongly associated with hemorrhagic colitis and HUS compared to Stx1 or to the presence of both genes [22,23,62,63]. Within the Stx2 type, Stx2a (formerly named Stx2), Stx2c, and Stx2d_activable_ [64] are associated with a higher risk for human disease [22,65]. Conversely, Stx2e is mostly associated with pig edema disease [66], and Stx2f was first isolated from the feces of feral pigeons [67] and, until recently, rarely reported in human illness [68,69,70,71]. Occurrences of Stx2e or Stx2f in human disease are thought to be extremely rare [72]. Nevertheless, Stx2f-producing EHEC infections are more common than expected [73,74,75].

## 3. STEC-HUS as a Zoonosis: Reservoirs, Sources, and Modes of Transmission

The importance of cattle as the primary reservoir for STEC has been hypothesized since the first outbreaks associated with undercooked hamburgers [17]. Occasionally, sheep [76] or goats [77] have been reported as sources of outbreaks. Cattle are asymptomatic carriers of STEC: after internalization in bovine epithelial cells, Shiga toxin is excluded from the endoplasmic reticulum and localizes to lysosomes, where its cytotoxicity is abrogated [78]. Reported prevalence in farm and slaughterhouse studies varies widely, but a recent meta-analysis yielded an estimated prevalence of *E. coli* O157:H7 in North America of 10.68% (95% CI: 9.17%–12.28%) in fed beef, 4.65% (95% CI: 3.37%–6.10%) in adult beef, and 1.79% (95% CI: 1.20%–2.48%) in adult dairy. In winter months, the prevalence was nearly 50% lower than that recorded in the summer months [79], consistent with the seasonality observed in human infections [80]. Contamination by EHEC decreases during processing of the meat [81], but some authors reported that salt at concentrations used for this process may in fact enhance Stx production [82]. Among animals positive for STEC, the term “super-shedder” is applied to cattle that shed concentrations of *E. coli* O157:H7 ≥ 10⁴ colony-forming units/g feces. This population of animals, which includes calves after weaning, could be responsible for the spread of the pathogen in the hide and the environment and, therefore, represent potential targets for veterinary interventions such as vaccination, bacteriophage therapy, probiotics, or dietary measures [83]. No difference was observed between organic and conventional farms [84], but antibiotic growth promoters may contribute to the expansion of STEC by triggering the bacterial SOS (see Individual Level in Section 6.1.2) response system [85]. Transmission to humans may occur through various routes: consumption of meat and dairy products (foodborne), contamination of crops or drinking water (waterborne) by animal waste, or direct person to person transmission due to a very low infective dose [86]. Rarely, transmission from cattle to farmer has been implicated [87]. The role of ground beef as a vehicle for STEC seems to be decreasing, and recent outbreaks have been associated with raw milk products, spinach [88], municipal drinking water [89], or fenugreek [90]. In a retrospective analysis of 350 outbreaks in the USA between 1982 and 2002, Rangel and colleagues found that 52% of outbreaks were foodborne (including 21% for which ground beef was the transmission route), 14% resulted from person to person transmission, and 6% from recreational water. The transmission route remained unknown after investigation in 21% of outbreaks [91]. It is noteworthy that *E. coli* can survive for months in the environment, potentially leading to the contamination of fresh produce [92].

### 3.1. Global Burden, Spatial and Temporal Distribution of STEC-HUS Cases

Hemorrhagic colitis and STEC-HUS represent serious health issues, although the global burden remains unclear, chiefly because of the lack of diagnostic tools that are easy to use in routine and a loose surveillance network in many countries. Nonetheless, it has been estimated that STEC accounts for 2.8 million acute illnesses and 3890 HUS cases annually [8], with a slight decrease in its incidence since 2000 [8,93]. The estimated cost of STEC-associated diseases could exceed US$400 million [94,95]. STEC-HUS is one of the most common diseases requiring emergency renal replacement therapy in children [96] and is responsible for 2%–5% of mortality worldwide during the acute phase [97]. In global terms, the incidence of STEC-associated diseases varies widely, mainly in relation to environmental and agricultural factors such as stockbreeding, with Argentina having the highest prevalence worldwide: 12.2 cases per 100,000 children younger than 5 years old, approximately 10-fold higher than that in other industrialized countries [8,98,99]. New Zealand reports an annual infection rate of 3.3 per 100,000 persons, whereas neighboring Australia only reports 0.4 cases per 100,000 persons [100]. Rural areas also tend to be more affected than urban ones [101,102], and cases occur predominantly during summer months [99,103]. Incidence rates of HUS vary greatly depending on the age of the patient and have peaked to 3.3 cases per 100,000 children-years in children aged 6 months to 2 years, for example in France [80]. Contrary to common belief, most cases of STEC-HUS are sporadic [99,104], and the incidence of STEC-HUS has been fairly steady since its recognition in the 1980s, with only a slight decrease after 2000 [8], despite public and industry efforts to reduce the risk of food and water contamination [105].

### 3.2. Propensity to Develop STEC-HUS

Approximately 5%–10% of infected patients will develop STEC-HUS about a week after the onset of digestive signs. The propensity to develop the disease varies according to microbiological and individual characteristics, although the determinants of the disease are not fully elucidated. First, the risk of HUS is greater for O157:H7 *E. coli* (≈10%) and Stx2v-harboring strains than for non-O157 serotypes and Stx1-harboring strains (≈1%) [22,23,41,43,50,62,63]. Since the first documented outbreaks in the 1980s, the O157:H7 strain has genetically diversified and concurrently acquired enhanced virulence due to bacteriophage-related insertions, deletions, and duplications [106]. Second, age is also an important risk factor for HUS, with peak incidence below 5 years and above 65 years [99,103,107]. Gastric acidity is an important barrier to ingested pathogens, and the use of anti-acid medications has been suggested as a risk factor [93]. Behavioral and environmental factors such as eating undercooked meat, contact with farm animals, and consumption of raw milk or well water have been described as risk factors in case control studies [93]. Some authors also reported that female sex [108] and a higher socio-economic status [109] are associated with a higher risk of developing STEC-related disease. Genetic factors, like erythrocyte and serum Gb3 level [110,111] or presence of the platelet glycoprotein 1b alpha 145M allele [112], could also influence the susceptibility to HUS. 

## 4. Pathogenesis

EHEC ranks among the most dreaded enteric pathogens in temperate countries. Following ingestion of contaminated food or water, EHEC displays a sophisticated molecular machinery consisting of a dual strategy: colonization of the bowel and Shiga toxin production. Recent progress in the understanding of HUS mechanisms has highlighted the role of the complement pathway in endothelial damage and gone a long way in deciphering the intracellular trafficking of Shiga toxin. However, most studies have focused on the O157:H7 serotype, and whether the mechanisms uncovered in the setting of O157:H7 infections apply to non-O157 strains, or whether specific mechanisms are involved, is speculative. Another shortcoming has long been the absence of a reliable animal model. Briefly, until recently, murine models did not fully recapitulate the features of STEC-HUS as a result of predominant expression of Gb3 on mouse tubular cells [113], as opposed to glomerular endothelial cells in humans [114,115]. Previous mouse models also relied on the co-injection of lipopolysaccharide (LPS) in order to boost cytotoxicity [116,117,118,119,120], thus obscuring the significance of the results considering the uncertainty regarding the implication of LPS in this pathology in humans. Indeed, even though the LPS-binding protein has been reported to be elevated in STEC-HUS patients [121], the role of endotoxinemia has never been properly demonstrated, as opposed to HUS resulting from Shigellosis [122]. More recently, new models have been created with refined Stx2 injection strategies and without the need for LPS injections. These models exhibit a wider range of the pathomechanisms expected in HUS [123]. Primate models have sometimes provided conflicting results, as exemplified in Section 4.4 in complement pathway research.

### 4.1. Colonization of the Bowel: The Attaching and Effacing Phenotype

Prior to adhering to the enterocytes, EHEC must first penetrate the thick mucus layer that protects the enterocytes. It accomplishes this by secreting the StcE metalloprotease, which reduces the inner mucus layer, thus allowing EHEC to access the intestinal epithelium [124]. Like the enteropathogenic pathovar [20], typical EHEC harbor the LEE [125], which includes the type 3 secretion system, a protein appendage capable of translocating a wide repertoire of effector proteins into the cytoplasm of the target cell in the distal ileum. Among these, the translocated intimin receptor (Tir), once injected into the host cell, allows for the attachment of the bacterium. Once expressed on the surface of the enterocyte it acts as the receptor for intimin (*eae*) and consolidates the attachment of *E. coli* to mucosal surfaces initiated by their flagella [126] and pili [127]. Stx plays a role in reinforcing *E. coli* adherence to the epithelium by increasing the expression of nucleolin, another surface receptor for intimin [128]. Tir also links the extracellular bacterium to the cytoskeleton of the host cell via a Tir-cytoskeleton coupling protein (Tccp, also known as EspF(U)), in the presence of a host protein insulin receptor substrate protein of 53 kDa (IRSp53) [129], in a process termed “pedestal formation”. Tccp, in turn, activates the actin nucleation-promoting factor WASP/N-WASP, enabling *E. coli* to literally seize control of the eukaryotic cytoskeletal machinery [130,131]. However, EHEC are not tissue invasive and, if it was not due to Shiga toxins, their pathological effect would be identical to enteropathogenic *E. coli* (i.e., invasion of the colon, disruption of tight junctions, and effacement of microvilli, resulting in watery diarrhea) [20].

### 4.2. Shiga Toxin Production and Effect: Gb3 Fixation and Trafficking

After bacterial lysis, Shiga toxins are released into the intestinal lumen, and its B subunit binds to its receptor globotriaosylceramide (Gb3) (see Section 2.2). Normal enterocytes (at variance with colon cancer cells [132]) do not express Gb3. Thus, it is believed that Stx translocates across the intestinal epithelium tight junction by binding to Gb3 expressed on Paneth cells, which are seated in the deep crypts of the small intestine [133]. Stx does so either by paracellular transport during neutrophil (PMN) transmigration, or by Gb3-independent transcytosis and macropinocytosis [134,135] before being released into the bloodstream. The mechanisms governing the circulation of Stx from the intestines to the target organs are still debated (reviewed in [136]). Some authors point to the role of polymorphonuclear leucocytes as potential carriers [137,138], but these results have yet to be replicated [139,140], and Stx possibly only binds to mature polymorphonuclear cells [141]. In any case, the estimated half-life of Stx in serum is less than 5 min, as it rapidly diffuses to affected tissues [142]. It is thus likely that by the time patients develop HUS, Stx has disappeared from the serum [140,143]. Expression of Gb3 in humans is restricted to podocytes, microvascular endothelial cells (the highest content being found on microvascular glomeruli) [144,145], platelets [146], germinal center B lymphocytes [147], erythrocytes (where it constitutes the rare P^k^ antigen), and neurons [148]. The physiological role of this glycosphingolipid and the reasons behind its specific distribution in human tissues are unknown. A Gb3 knock-out mouse model resulted in no apparent phenotype, except for the loss of sensitivity to Shiga toxins [149]. In Gb3-positive cells, the Stx-Gb3 complex induces membrane invagination [150] that facilitates endocytosis. Importantly, this initial process of Stx endocytosis is highly dependent on the close connection of Gb3 and lipid rafts [151] stemming from animal cell membranes. Indeed, lipid rafts contain caveolin where polymerization provides the platform on which to form early endosomes. The mobilization of microtubular units bring into play both clathrin-dependent [150,152] and clathrin-independent pathways [153,154]. Next, the Stx-Gb3 complex is addressed from early endosomes to the endoplasmic reticulum though retrograde transport, making it possible for Stx Gb3 to escape lysosomal degradation [155]. During transport [155], the catalytic A subunit is cleaved by the protease furin into two fragments: A1 and A2. In the endoplasmic reticulum, the disulfide bound between the two fragments is reduced [156], and the A1 fragment translocates into the cytoplasm (anterograde transport) where it is free to exert its cytotoxic effects by removing an adenine base at the *N*-glycosidic bond from the 28S rRNA of the 60S ribosome [157], thus inhibiting protein synthesis leading to cell death [57,158]. The mechanism allowing Shiga toxins to bypass late endosomes and lysosomes is only partially known, but is thought to involve cycling Golgi protein GPP130, which is susceptible to degradation by physiological concentrations of manganese, yielding hope for a future therapeutic application [159]. The pathophysiology of Shiga toxin trafficking and intracellular action is schematized in Figure 3.

### 4.3. Mechanisms of Shiga Toxin Cytotoxicity

Inhibition of protein translation by ribotoxic stress is the prominent mechanism of Stx cytotoxicity and a major gateway to apoptosis. The processed A1 fragment cleaves one adenine residue from the 28S RNA of the 60S ribosomal subunit, thus inhibiting protein translation and triggering the ribotoxic and endoplasmic reticulum stress responses, which in turn paves the way for cell apoptosis through p38 mitogen-activated protein kinase (p38 MAPK) activation [160,161] and various apoptotic pathways depending on the infected cell type [162]. In addition to its ribotoxic effect, Shiga toxin activates multiple stress signaling and apoptotic pathways, and it is responsible for the production of inflammatory cytokines by target cells. On the cell surface of monocytes, Gb3 surface expression is not associated with lipid rafts, which means that Stx is routed towards the lysosomal pathway [163,164,165]. This results in the production of a high amount of TNF-α, GM-CSF, and IL-8 by monocytes in response to Stx, enhancing endothelial dysfunction and organ damage in patients with HUS [166,167,168,169] along a ribotoxic-independent route [170]. Stx can also be found in Gb3-negative intestinal cells (probably after internalization by macropinocytosis/transcytosis), where it can modulate the immune response by inhibiting the PI3K/NF-ϰB pathway [171].

### 4.4. Activation of Complement Pathways: Culprit or Innocent Bystander?

By unraveling the role of alternative pathway dysregulation in atypical HUS [172,173,174,175,176,177], investigators have initiated the use of eculizumab, a terminal C5 inhibitor, which is now established as a mainstay in the management of patients with atypical HUS [178,179,180,181,182]. Evidence has also been garnered suggesting the participation of an alternative pathway in STEC-HUS [183]. Plasma levels of Bb and C5b-9, two complement pathway products [184], and C3-bearing microparticles from platelets and monocytes [185,186], were found to be elevated in patients suffering from STEC-HUS. Both decreased at recovery but were not associated with disease severity. Recent in vitro studies demonstrated that Stx is capable of directly activating complement, in addition to its cytotoxic effects. Stx2 binds to complement factor H and its regulators [187,188]. Furthermore, Stx2 induces the expression of P-selectin on the human microvascular endothelial cell surface, which binds and activates C3 via the alternative pathway, leading to thrombi formation in a murine model of STEC-HUS [189]. Recently, serological and genetic complement alterations were reported in 28% of STEC-HUS children [190]. Nevertheless, these intriguing results have been diminished by the inability to replicate the findings in nonhuman primate models [191]. The absence of C4d or C5b9 by immunochemistry in biopsies from 11 patients during the O104:H4 outbreak is also a source of concern [192]. Lastly, mice lacking the lectin-like domain of *thrombomodulin*, an endothelial glycoprotein with anticoagulant, anti-inflammatory, and cytoprotective properties, show higher glomerular C3 deposits and a higher mortality after intraperitoneal injection of Stx2 + LPS [193], and a deficiency of this protein has been implied in rare cases of atypical HUS [194]. Although preliminary, these results could provide the rationale for the use of ART-123, a human recombinant thrombomodulin tested in the setting of disseminated intravascular coagulation (without improvement of all-cause mortality) [195] and acute exacerbations of idiopathic pulmonary fibrosis (ongoing, NCT02739165), in STEC-HUS. Published results from three patients are encouraging [196].

### 4.5. Endothelial Damage: From Stx Cytotoxicity to Thrombotic Microangiopathy

Once released into the bloodstream, Stx reach target organs [197] and bind Gb3 on microvascular endothelial cells. Differences in Gb3 expression distribution across various vascular beds are the basis for differential organ susceptibility to Stx [144,198,199]. Vascular dysfunction is both a hallmark of Shiga toxin pathophysiology and an early harbinger of negative clinical outcomes [200,201]. Damage to the vascular bed can broadly be categorized as (1) direct cytotoxicity to the endothelium; (2) disturbance of the hemostatic pathway; (3) enhanced release of chemokines; and (4) alternative pathway activation [198,201]. Shiga toxins induce a profound remodeling of the gene expression repertoire of endothelial cells rather than prompting cell death, provided that vascular cells are subjected to sublethal concentrations of Shiga toxin [199,202]. The net effect is that endothelial cells adopt a prothrombogenic phenotype by expressing increased levels of tissue factor (TF) [203], releasing augmented levels of von Willebrand factor [204,205], and activating platelets [206] via the CXCR4/CXCR7/SDF-1 pathway [202]. In addition, Stx stimulates the expression of adhesion molecules [207] and inflammatory chemokines [208], thereby potentiating the cytotoxity of Stx [209] and promoting the adhesion of leucocytes to endothelial cells, which in turn exacerbate thrombosis and tissue damage. At higher concentrations, Stx trigger endothelial apoptosis and cell detachment, exposing the subendothelial bed rich with prothrombogenic tissue factor and collagen [209,210,211]. Finally, Stx elicits the formation of C3- and/or C9-coated microvesicles derived from platelets or red blood cells [185,186,212]. Complement fraction C3a is believed to activate microvascular thrombosis by mobilizing P-selectin on the surface of endothelial cells [189]. As a result, Stx-mediated changes in the endothelial phenotype result in a prothrombogenic environment, demonstrated by higher median plasma concentrations of prothrombin fragments, tissue plasminogen activator (t-PA), and D-dimer in children in whom STEC-HUS develops, compared to those with uncomplicated infection [200].

## 5. Diagnosis

### 5.1. Clinical Presentation

STEC-related diseases display a wide range of severity, from asymptomatic carriage to lethal HUS. Rapid identification of symptoms compatible with EHEC infection is indispensable, both to allow for appropriate patient care and for epidemic control. Yet, clinicians are often bewildered by the misleading clinical presentations of STEC-HUS, especially in adults, and deceived by misconceptions regarding the disease (Table 1), namely that it essentially occurs as part of large outbreaks and that ground beef represents its main vector [213,214]. Importantly, most investigations have focused on the clinical presentation in children. However, symptoms at presentation can differ greatly with age [215] and, with the exception of a few large outbreaks that have been the subject of extensive study [9,216], there is a dearth of data regarding the clinical presentation in adults.

### 5.2. From Colitis to HUS

The proportion of patients exposed to EHEC who will develop colitis (attack rate) varies considerably, from 14% [222] to 33% [216], depending on individual (age) [99,103,107], immunological factors, and strain characteristics [41]. The infective dose is probably very low, with less than one *E. coli* O111 per 10 g of fermented sausage in the 1996 Australian outbreak [223] and a median 68 *E. coli* O157:H7 per hamburger patty in the 1993 outbreak in western USA [224]. After a median incubation of 4 (1 to 10) d [225,226] following ingestion of the inoculum, patients usually present with painful diarrhea and abdominal cramping. Vomiting (20%–30%) and fever (10%–40%) are less frequent, the disease being usually limited to the colon and not prone to bacteremia. Bloody diarrhea only occurs in a second stage, between one to five days after the onset of the symptoms [227]. Of note, bloody diarrhea is not a defining feature of STEC-HUS and it may never occur in 20%–30% of patients [41,218]. Exceptionally, hemorrhagic colitis can be severe and necessitate bowel resection [228], or result in rectal prolapse [229]. Patients infected with non-O157 EHEC usually have a milder disease severity, and 95%–99% will heal spontaneously within seven days [230]. In contrast, the proportion of patients whose course is complicated by HUS is approximately 10% for O157:H7 infection [41,91], and once again hinges on the characteristics of both patient and strain (see Section 2.1.3). Timeframe and evolution from colitis to STEC-HUS, along with the theoretical window for diagnostic tests, are depicted in Figure 4.

### 5.3. Clinical Predictors of Evolution Towards HUS

The lack of factors that reliably predict the occurrence of HUS is still a significant obstacle for clinicians facing patients infected with EHEC. Discrepancies between studies can be explained by inter alia strain variability, that is, a predictor identified during a specific outbreak may not necessarily be relevant in other cases. Clinical predictors include dehydration [231,232], fever [233], vomiting [221,234], visible blood in the stool, older [234] or younger age [235], and use of antimotility agents in the first three days of illness [236]. The use of certain antibiotics could also be associated with the development of STEC-HUS (see Individual Level in Section 6.1.2).

### 5.4. Renal Involvement

Acute kidney injury (AKI) in STEC-HUS patients ranges from asymptomatic urine sediment abnormalities to severe renal failure and end-stage renal disease. Proteinuria is usually mild and has been described in 30% of patients, combined with hematuria in 6.6% and leukocyturia in 26% [237]. Between 30% [221] and 61% [218,238] of STEC-HUS patients require renal replacement therapy (RRT) during the course of the disease, with a mean duration of oliguria or RRT of 9–10 d [239,240], and 15% of children develop hypertension [237,238]. In addition to blood urea nitrogen and creatinine levels, neutrophil gelatinase-associated lipocalin (NGAL) could be a useful biomarker for the diagnosis of acute kidney injury and in predicting the need for RRT [241].

### 5.5. Extra-Renal Involvement

The kidney and the brain are the organs most vulnerable to STEC-HUS [242], but several other organ involvements have been described and need to be considered when evaluating patients with STEC-HUS. 

#### 5.5.1. Neurologic Involvement

Neurologic involvement is one of the most dreaded complications of STEC-HUS. It is responsible for the majority of patient deaths [243] and is an important contributor to the morbidity of the disease. Approximately 25% [238] of STEC-HUS patients develop neurologic symptoms after a median delay of four days following the onset of HUS [244]. The two most common neurologic manifestations are coma and seizures, but various focal defects, pyramidal or extrapyramidal syndromes have been described [245,246,247]. During the O104:H4 outbreak, in which half of the patients developed neurologic symptoms, epileptic seizures were seen in 20% and cognitive impairment or aphasia in 67.3% [244]. In addition, older patients are prone to psychiatric symptoms [248]. Fatal outcome is recorded in around 20% of patients with neurologic involvement, and severe sequelae is observed in about 27% of these patients [245,246]. Magnetic resonance imaging (MRI) and histopathological studies have pointed out that basically every structure of the central nervous system can be affected, consistent with the ubiquitous distribution of Gb3 in neurons [148], although astrogliosis and microgliosis are especially prominent in the thalamus and the cortex [244]. Multiple resonance imaging with apparent diffusion coefficient is almost always abnormal when patients present with neurologic symptoms, but these early findings do not seem to reinforce clinical prediction nor to correlate with symptoms [249,250]. Lastly, neurologic complications often parallel renal failure and are exceedingly rare in the absence of AKI. Neurologic symptoms as the unique manifestation of thrombotic microangiopathy should prompt clinicians to consider the diagnosis of thrombotic thrombocytopenic purpura (TTP) rather than STEC-HUS.

#### 5.5.2. Cardiac Involvement

Although rarer, acute myocardial infarction is another potentially life-threatening complication of STEC-HUS [251]. Its incidence has not been properly evaluated, and even though histologic lesions have been identified in 30% of autopsied cases [252], clinical manifestations (cardiac ischemia, rhythm disorders, cardiac arrest) seem to occur in less than 10% of STEC-HUS pediatric patients [253,254,255]. Pericardial involvement with cardiac tamponade has also been recorded [256].

#### 5.5.3. STEC-HUS and Diabetes Mellitus

Biological pancreatitis, as well as elevated liver enzymes, occur in 20% of STEC-HUS patients [237] but do not commonly result in organ failure. Nevertheless, around 3% of patients have hyperglycemia during the acute phase [257], and survivors of STEC-HUS (but not uncomplicated EHEC infection) have a significantly increased incidence of diabetes [258], possibly as a consequence of thrombosis of vessels supplying the islets of Langerhans as evidenced in autopsy series. Diabetes may be transient, yet the partial reduction in the stock of Langerhans islets may translate to the re-emergence of diabetes after a variable delay [252,258,259].

### 5.6. Recurrence

STEC-HUS does not usually recur, and a second bout of the disease should lead to the suspicion of alternative complement pathway dysregulation [260]. Patients develop antibodies that may, in part, be protective [101], but in the case of repeated exposition to Stx, recurrence has been described [261,262], for example with the atypical O80 serogroup [46].

### 5.7. Unusual Invasive Infections 

Unusual extra-intestinal infections such as bacteremia or deep abscesses have recently been described for the emerging O80 serogroup EHEC, whereas EHEC is generally known to be a strictly intestinal pathogen [45,46]. Other rare cases of bacteremia due to EHEC strains have been described, in particular following urinary tract infections [263,264,265,266,267]. Nevertheless, for the O80 serogroup, additional extra-intestinal VFs characteristic of the plasmid pS88 are consistently associated with classical EHEC intestinal VF (*eae*, *stx*, *ehxA*). pS88 is a key determinant of extraintestinal *E. coli* (ExPEC) virulence, involved in neonatal meningitis [268,269].

### 5.8. Paraclinical Signs

#### 5.8.1. Thrombotic Microangiopathy

HUS is a type of thrombotic microangiopathy and is, therefore, defined by the triad of Coombs-negative anemia with erythrocyte fragmentation (as seen on a peripheral blood smear by the presence of schistocytes), thrombocytopenia, and ischemic organ failure [1]. Anemia is often severe and sudden, requiring red blood cell transfusion in more than 80% of cases. Thrombocytopenia can also be profound, with a reported mean nadir of 37 G/L [238], and the risk of platelet transfusion is discussed in Section 6.2.4. Other features of hemolysis include elevated Lactate deshydrogenase (LDH) elevated indirect bilirubin levels, and undetectable haptoglobin. The severity of thrombocytopenia does not correlate with kidney injury or outcome, but median peak LDH levels are higher in patients requiring RRT, and these patients require more red blood cell transfusions [270]. Along with the decrease in LDH, the rise of the platelet count is one of the first signs signifying recovery from STEC-HUS, usually within 10 to 14 d after disease onset. At variance, anemia can persist for a longer period, in particular in the presence of prolonged renal failure, and may necessitate the prescription of erythropoiesis stimulating agents (ESA) to alleviate blood transfusion requirements [271,272].

#### 5.8.2. Inflammatory Features and Coagulation Activation

STEC-HUS patients display inflammatory features with elevated leucocyte count (often more than 15 G/L [238,273]), C-reactive protein, and fibrinogen. Plasma concentrations of prothrombin, fragment 1+2 tissue plasminogen activator (t-PA) antigen, t-PA–plasminogen-activator inhibitor type 1 (PAI-1) complex, and D-dimer are also elevated, providing further evidence of the disequilibrium between enhanced thrombin generation and inhibited fibrinolysis. These prothrombotic coagulation markers precede HUS and may, therefore, herald its occurrence [200].

#### 5.8.3. Biological Predictors of Evolution Towards HUS

Along with clinical predictors (Section 5.3), the degree of systemic inflammation seems to correlate with the evolution from colitis to STEC-HUS. In particular, higher leucocyte count [221,233,235,236] and C-reactive protein level >1.2 mg/dL [233] are independent risk factors for STEC-HUS, consistent with the role of cytokine production in the development of the disease. Proteinuria has also been occasionally described as associated with the onset of STEC-HUS [235].

#### 5.8.4. Histopathology

Renal biopsy is only performed in the context of STEC-HUS in the case of diagnostic uncertainty, which makes its histopathological description rare and potentially biased. Nevertheless, STEC-HUS patients seem to display unspecific features of thrombotic microangiopathy, such as glomerular capillary thromboses with a widened subendothelial space, endothelial swelling, and congested glomeruli. Necrosis of capillary walls, with luminal narrowing and thrombosis, is also characteristic. Cortical infarcts can be seen in severe and fatal cases [115], and superimposed acute tubular damage is commonplace [192]. Immunochemistry for C1q, C3, C4d, or C5b-9 does not seem to detect complement deposition in the glomeruli [192]. 

### 5.9. Microbiology

A prompt and accurate etiological diagnosis is needed in the face of a thrombotic microangiopathy syndrome in order to tailor the initial treatment that is specific to each etiology [1]. Likewise, during a diarrhea outbreak, rapid identification of EHEC allows for timely epidemiological investigations and isolation measures that will prevent further spreading of the pathogen, as well as avoidance of antibiotic therapy and antimotility agents in cases of STEC-related disease [214]. Early clinical and biological signs do not easily permit the distinction of STEC-HUS from other thrombotic microangiopathy syndromes or STEC-associated colitis from enteropathogenic agent colitis (see Section 4.4). Thus, the importance of a rapid diagnosis relies on microbiological tools, first of all on detection of Shiga toxin by molecular diagnosis or immunoassay. Isolation of a Shiga toxin-producing *E. coli* is also crucial for epidemiological surveillance, such as within the PulseNet International network [274]. Therefore, current US guidelines recommend plating stools from patients with acute community-acquired diarrhea on a selective medium, in combination with an assay that detects the Shiga toxins or the genes encoding these toxins [219]. In addition, every patient presenting with a thrombotic microangiopathy syndrome, irrespective of the presence of inaugural bloody diarrhea or neurologic symptoms, should be investigated for Shiga toxin and STEC. A sequential two-step strategy with a non-culture assay, followed by culture for STEC in the event of positive Shiga toxin, represent an unacceptable delay for the isolation of STEC strains. Selective testing on the basis of a patient’s age or season of the year is also inappropriate, and it has been shown that the prevalence of Stx was identical whether routine screening is implemented or if the analysis is based on physician’s request [63]. In 2000, 68% of US clinical laboratories reported routinely testing stool specimens for *E. coli* O157:H7 with stool culture, an immunoassay, or both [275]. Shedding of EHEC is transient, and its isolation is highly dependent on obtaining stool cultures or rectal swabs within six days of onset of diarrhea [276], prior to any antibiotic therapy. The amount of free fecal Shiga toxin is low, and the likelihood of identifying Shiga toxin decreases dramatically over the time course of the disease [277]. Factors associated with success in identifying STEC in pediatric post-diarrheal HUS include testing less than 4 d after onset of symptoms, patient age older than 12 months, cases related to an outbreak of STEC-HUS, patients presenting with bloody diarrhea during the summer months, high leucocyte count, and moderate anemia [220].

#### 5.9.1. Identification of EHEC: Culture and Characterization

*E. coli* O157:H7 lost its capacity to ferment sorbitol during its evolution [27]. Therefore, culture on a sorbitol-containing selective medium, such as sorbitol–MacConkey agar (SMAC) [278], facilitates identification of *E. coli* O157:H7, which appears colorless after incubation for 16 to 24 h. The addition of cefixime tellurite (CT-SMAC) or bile salts can suppress the growth of irrelevant flora and increase the sensitivity of the culture [279]. Other species can occasionally carry the O157 antigen, and confirmation that the isolated colony consists of *E. coli* is warranted [280]. However, sorbitol fermenting O157 and non-O157 strains go undetected on the McConkey agar medium. The use of a chromogenic medium has recently been designed for detecting both O157 and non-O157 STEC from clinical samples [281]. Sorbitol fermenting O157 is a pathogen of great virulence and has been repeatedly incriminated in deadly outbreaks across Europe [37,38,282]. This fact combined with the low yield of stool cultures after four days [220,283,284] makes a strong case for the concurrent use of nonculture-based assays.

#### 5.9.2. Identification of Shiga Toxin: Non-Culture Assays

The emergence of non-culture assays has facilitated the diagnosis of STEC-HUS and highlighted the importance of non-O157 strains in the epidemiology of STEC-related diseases. Non-culture assays are quicker and have the potential to detect all serotypes of EHEC potentially involved in STEC-related diseases. Its main drawback is that the infective organism is not isolated, thus restricting the clinical relevance of the result and the potential for public health interventions. Another pitfall when relying exclusively on assays detecting Shiga toxins is the potential loss of toxin production by EHEC during the course of the infection, which significantly hampers the sensitivity of this technique in the later stages of the disease [285,286].

##### Molecular Biology 

Diagnosis relies on the detection and distinction of genes encoding Shiga toxins (*stx1* and/or *stx2*) by polymerase chain reaction (PCR) following stool enrichment to maximize the sensitivity. The results are then available within 12 to 24 h. Depending on the primer used, PCR can also detect *stx* subtypes; virulence-associated genes such as *eae* encoding for intimin, *ehxA* encoding enterohemolysin, and *aggR* encoding for aggregative adherence fimbria I; or the specific O group of the pathogen [287]. All these features may detect risk factors for HUS evolution. Multiplex PCR [288] and real-time PCR [254] have been developed and allow an earlier diagnosis (less than 24 h) compared to traditional methods. PCR found Stx in the stools of infected patients during a median of 20 d (1–256 d) after onset of symptoms [63]. Early isolation and characterization of STEC strains enable epidemiological surveillance and cluster detection by performing molecular analyses such as whole-genome sequencing [289]. The determination of serotype (O and H antigens), virulence genes (*stx* and their subtypes *eae*, *ehxA*, *saa*, *aggR* and *subA* genes), acquired resistance genes, and multilocus sequence typing (MLST) are performed using tools available at the Center for Genomic Epidemiology (https://cge.cbs.dtu.dk/services/) [290]. Phylogenetic analysis is performed by single nucleotide polymorphisms (SNPs), and a core genome MLST (cgMLST) analysis is integrated into Enterobase [291]. These tools are crucial to rapidly detect clusters of STEC strains and, thus, to identify outbreaks and take preventive measures. 

##### Immunological Tests

Immunoassays (reviewed in [219,292]) are based on enzymatic, optical, magnetic, or immunochromatographic tests to detect Shiga toxin 1, 2 or both. All necessitate overnight enrichment in broth cultures. Comparisons of the diagnostic performances of the different immunoassays available are currently lacking, but sensitivity is usually lower compared to PCR [284,293,294], and a negative result in the presence of strong clinical suspicion of HUS requires confirmation using the PCR method.

##### Serodiagnosis

Serodiagnosis can be useful in cases where isolation of EHEC or Shiga toxin could not be performed, or was negative despite strong clinical suspicion, but is barely made at present, except for a few serotypes and with poor discriminative value. Detection of antibodies directed against LPS (O-groups) seems to be of greatest diagnostic value: IgM appear soon after the infection and peak at day 9, whereas IgG appear from day 8 [295] and persist several weeks after infection [296]. Repeated serology after two to three weeks may demonstrate an increase in antibody titers. The combination of serology with standard fecal diagnostic tests could be specifically useful when patients present late in the course of the disease and at the time of HUS [283], or for epidemiological purposes [101]. Important caveats remain about the possibility of cross-reactions with other bacterial strains belonging to different genera (*Salmonella, Yersinia, Citrobacter*), with which *E. coli* O157 shares epitopes, and in the lack of sensitivity for non-O157 serotypes.

### 5.10. Differential Diagnosis

At the early stage of acute bloody diarrhea, Shiga toxin-producing *E. coli* is hardly distinguishable from other pathogens (*Campylobacter, Salmonella, Yersinia, Shigella, Clostridium difficile,* and other pathogenic serovars of *E. coli*) and non-infectious causes (appendicitis, intussusception, colorectal cancer, and ulcerative and ischemic colitis) based on the general clinical and biological criteria [297]. Fever is rare, compared to entero-invasive pathogens, and occurrence of diarrhea during hospitalization and antibiotic therapy are atypical. Patient history should include recent travels and food consumption. Computed tomography can help to rule out ischemic colitis in adults [298], but the added value in acute bloody diarrhea is otherwise scant, and imaging studies are not required [299]. On the contrary, prompt investigations for STEC and Shiga toxin in patients with community-acquired diarrhea are mandatory. In the pediatric setting, STEC-HUS represents the vast majority of thrombotic microangiopathy cases (>80%) [2], and additional biological testing, such as monitoring the protease ADAMTS13 activity or complement investigations, are mostly decided depending on atypical clinical presentation or after exclusion of STEC infection. In adults, the initial etiological approach should be grounded in the patient’s clinical setting and existing conditions; bone marrow or solid organ transplantation, drugs, HIV infection, malignant hypertension, or metastatic malignancy suggest a secondary thrombotic microangiopathy syndrome. The next step consists of discriminating STEC-HUS from atypical HUS, and TTP and represents a far greater challenge if only for the inconsistent presence of hemorrhagic diarrhea that is found in a substantial proportion of patients with TTP and atypical HUS [300]. Dysregulation of the alternative complement pathway, ADAMTS13 activity, and the presence of Shiga toxin-producing *E. coli* should be examined in every patient, bearing in mind that these investigations require delays irreconcilable with the necessity for prompt targeted therapies. Most studies, to date, have focused on identifying features that differentiate TTP and HUS, on one hand [301,302,303,304], or TTP with other thrombotic microangiopathies on the other [305]. TTP patients typically display lower platelet counts, higher reticulocyte count, and lower creatinine and blood urea nitrogen levels. The presence of antinuclear antibodies provides another clue for diagnosis. Clinical and general biological features do not reliably distinguish STEC-related from atypical HUS [300]; therefore, stool culture combined with an assay that detects Shiga toxins should be performed each time a patient presents with thrombotic microangiopathy.

## 6. Treatment

STEC-HUS stands at the crossroads between veterinary medicine, public health, and acute care, and this multidisciplinarity is reflected in its treatment, which implies veterinary and industrial preventive measures, epidemiological interventions when an outbreak occurs, as well as hospital-based and sometimes intensive care for STEC-HUS patients.

### 6.1. Prevention

#### 6.1.1. Primary Prevention

##### Individual Level

Hygiene: Modifiable individual risk factors for contamination with a STEC include consumption of raw beef, raw milk products, vegetables or sprouts [93], in particular for young children. Thus, proper hand [306] and food hygiene [307] are the main preventive measures. Cooking thoroughly, pasteurizing, or irradiating the food remove all EHEC, but supplementary precautions should be taken when visiting farms or for individuals working in contact with ruminants.

Vaccination: Several vaccines directed against the *E. coli* O157 LPS antigen [308] or Stx epitopes [309,310,311,312] have been validated in murine models and phase 2 studies, but none so far has proven its efficiency in reducing the risk of EHEC infection in humans [313].

##### Farm and Industry Level

Prevention of animal carriage, reviewed in [314], can be categorized as follows: (1) Animal vaccination, at variance with human vaccination, has proven its efficacy [315] in reducing the shedding of *E. coli* O157. (2) Some dietary manipulations, including probiotics [316], especially *Lactobacillus acidophilus,* and changing diet from grain to forage before slaughter [317] also decreases fecal *E. coli* in cattle. (3) Lastly, farm practices can be improved by providing dry bedding, keeping animals in the same herd groupings [318], and solarization of soil in feedlot pens [319]. 

##### Slaughterhouse Hygiene and Meat Processing

The importance of slaughterhouse hygiene and meat processing cannot be highlighted enough, considering a study published in 2000 reported a contamination by EHEC O157 in 87% of lots tested pre-evisceration, 57% post-evisceration, and this proportion dropped to 17% after processing (including antimicrobial treatment) [81]. The sanitary procedures implemented in slaughterhouses are, therefore, of tremendous importance. Guidelines for safe food handling and processing are provided by the United States Department of Agriculture Food Safety and Inspection Service [320].

#### 6.1.2. Secondary Prevention

##### Community Level

Once an outbreak is recognized, in addition to measures described in Section 6.1.1, exclusion from work or school and separation of pediatric patients from their siblings should be advised [321], whilst hospitalization of confirmed cases is recommended [214]. Recommendations summarized in the British guidelines [322] include identification of associated cases and vulnerable contacts and source-specific control measures. Prompt notification to public health authorities and recognition of the outbreak is of the utmost importance because it determines identification of the source. Outbreaks are recognized two or three weeks after contamination, and trading networks are becoming increasingly complex, which makes interviews and case control studies more challenging, as highlighted by the O104:H4 outbreak in Germany [90].

##### Individual Level 

Prevention measures also include interventions designed to impede the transition from colitis to HUS in infected patients.

Antimicrobial agents in the setting of HUS have sparked an ongoing controversy [323]. They were originally devised as a positive intervention to eliminate STEC, thereby diminishing Stx production and the risk of HUS. It was also argued that antibiotics could reduce fecal carriage of STEC and, thus, help to thwart the dissemination of the strain after early effective antimicrobial therapy for *S. dysenteriae* type 1 infection in Bangladesh [324]. Case control studies [107,221] and one prospective study [325] have dealt a blow to these arguments by connecting antibiotic therapy to the development of HUS. The debate was further fueled by two Japanese studies that reported a preventive effect of fosfomycin on the risk of STEC-HUS when given within the first three days of the illness [326,327]. Furthermore, during the O104:H4 outbreak, treatment with azithromycin was associated with a lower frequency of long-term STEC carriage in one center [328], and aggressive antibiotic treatment with ciprofloxacin and meropenem shortened STEC excretion in another center [329]. Treatment with azithromycin is currently being evaluated in a French clinical trial (ZITHROSHU, NCT02336516). Two meta-analyses [330,331] were performed to address this issue and concluded that antibiotics neither decreased nor increased the likelihood of STEC-HUS. However, a more recent meta-analysis [332] found that the association between antibiotic treatment and the risk of STEC-HUS did exist, provided studies at high risk for bias were excluded (12 out of 17). The single randomized trial designed to address this issue also found no significant effect of trimethoprim-sulfamethoxazole on progression of symptoms, fecal pathogen excretion, or the incidence of HUS, although its methodology has been called into question [333]. The explanation for these conflicting results lies, in great part, in the class-specific ability of certain antibiotics to induce phage replication and Shiga toxin release. Bacterial SOS response genes are expressed together with Stx phage genes, and fluoroquinolones, trimethoprim-sulfamethoxazole, and ß-lactams, which are SOS-inducing antimicrobial agents, are associated with Stx2 expression in vitro, whereas fosfomycin, rifampicin, gentamicin, doxycycline, erythromycin [33], and rifaximin [334] are not. In vivo models replicated these results with enhanced free fecal Shiga toxin and lethality in mouse or gnotobiotic piglet models of STEC-HUS treated with ciprofloxacin, and no effect of treatment was found with fosfomycin or azithromycin, respectively [335,336]. The response of *E. coli* O157 isolates to subinhibitory concentrations of antibiotics could also be dependent on the nature of the strain involved [337]. Clinical studies that segregated the role of specific antibiotic classes have demonstrated that ß-lactams [338], metronidazole, and trimethoprim-sulfamethoxazole [221] were associated with the most significant risk for HUS, whereas azithromycin [221] and aminoglycosides were protective against HUS development [338]. These results led some authors to advocate the use of antibiotic treatment with protein and cell wall synthesis inhibitors for STEC-HUS patients in specific situations [339]. Japanese guidelines for STEC-HUS and the French Haut Comité de Santé Publique (HCSP) [340] do not provide a definitive statement on the effectiveness of antibiotics in preventing HUS, but they consider treating asymptomatic carriers to prevent shedding of the pathogen [341]. In contrast, British guidelines [342] and the Infectious Disease Society of America (IDSA) caution against the use of antibiotics [343]. At any rate, their use should be weighed against the risk of aggravating the patient’s course through the induction of bacterial SOS response and Shiga toxin release and the potential toxicity in case of renal failure of dehydration. Despite evidence that antibiotics can aggravate the course of STEC-HUS, adult patients still receive unwarranted antimicrobial treatment in more than half of the cases, fluoroquinolones ranking as the most prescribed class. These results underscore the need for greater awareness among clinicians [344].

A small prospective cohort study showed in 2005 that the amount of sodium infused was associated with protection against developing oligoanuric HUS [231]. Since then, evidence has accumulated, and a recent meta-analysis demonstrated that a hematocrit value ≤ 23% was associated with an odds of 2.38 of developing oligoanuric renal failure [345]. Even though the risk of developing HUS was not assessed as an outcome variable per se, this potentially makes intravenous fluid expansion the first effective individual measure to prevent STEC-HUS and improve prognosis, and it represents a paradigm shift since, until recently, fluid restriction was the mainstay of treatment [346]. The hypothetic mechanisms of volume expansion consist in improving renal perfusion, counteracting the consequences of thrombi formation, avoiding ischemic organ damage, and maintaining tubular flow. Protocols are not well-established; most studies used isotonic saline, and the volume of intravenous fluid infused should be based on clinical assessment of intravascular volume in order to avoid fluid overload. Echocardiography or invasive hemodynamic monitoring in intensive care patients has not been evaluated in this context but may be of help in the hands of experienced clinicians. Monitoring of fluid intake and excreta is warranted.

During the O104:H4 outbreak, one center proposed daily intestinal lavage with polyethylene glycol (PEG) as a prevention for HUS in EHEC-infected patients, and they found this strategy to be efficient [347], but this finding remains to be validated on a wider scale.

### 6.2. Supportive Therapy

Supportive therapy is the cornerstone of the treatment of STEC-HUS patients once the disease is established, and it is mainly responsible for the improved prognosis in recent years [105]. Hospitalization is mandatory, preferably in specialized centers, and intensive care is often required [51]. Unspecific preventive and dialytic management of AKI will not be addressed in this review, and the reader is referred to the Kidney Disease: Improving Global Outcomes (KDIGO) guidelines [348], but specific measures, most of them relying on low grade evidence, are detailed below.

#### 6.2.1. Volume, Electrolytic Balance, and Nutrition

In addition to its protective effect on the development of oligoanuric renal failure in STEC-infected children (see Individual Level in Section 6.1.2), intravenous fluid expansion up to, and including, the day of STEC-HUS diagnosis has also proven to lessen the need for renal replacement therapy (RRT) (OR = 0.26, 95% CI 0.11–0.60) and reduce central nervous system-associated complications (OR = 0.26, 95% CI 0.07–0.91), as dehydration has been associated with mortality (OR = 5.13, 95% CI 1.50–17.57) [345]. Early volume expansion is therefore mandatory, but when AKI is established, it should be balanced against the risk of fluid overload. Nutrition, administered parenterally if necessary, is of special importance in toddlers and children [349], but its benefit may be extended to acutely ill patients of all ages [350].

#### 6.2.2. Blood Pressure Control

Hypertension is common in STEC-HUS patients, occurring in 15% of children [238], and is believed to result from fluid overload or renin–angiotensin system activation. No trial has ever compared the impact of assigning different blood pressure targets in STEC-HUS specifically, and thrombotic microangiopathies in general, but hypertension is a well-established contributor to thrombotic microangiopathy lesions [351] and could also partly account for the occurrence of posterior reversible encephalopathy syndrome (PRES) [352]. It should, therefore, be managed with appropriate medication, such as calcium receptor blockers or diuretics in the case of fluid overload. Angiotensin-converting enzyme inhibitors may be used, preferably after the acute phase [353].

#### 6.2.3. Renal Replacement Therapy (RRT)

Half of STEC-HUS patients will require RRT. If hemodialysis is the preferred renal replacement modality in adult patients, children are frequently treated with acute peritoneal dialysis. The choice of the method and its indication rely on centers’ protocols and general guidelines [348], and it has not been evaluated on a larger scale. In case of severe thrombocytopenia, regional citrate anticoagulation is advised [354,355].

#### 6.2.4. Transfusion

Packed red blood cell transfusion is required in most STEC-HUS patients [218]. Importantly, restrictive thresholds of 7 g/dL, advocated in the recent American Association of Blood Banks (AABB) guidelines, do not apply to patients with severe thrombocytopenia [356], and indications rely on individual patient characteristics and symptoms. Concerns regarding platelet transfusion in STEC-HUS are driven by the hypothetical fear that, by furnishing the cells needed for extensive microthrombi, platelet transfusion may exacerbate the disease. This concept is partly corroborated by data derived from other platelet-consumptive disorders such as heparin-induced thrombocytopenia [357] and TTP [358,359]. However, the risk has not been substantiated by two series of 77 pediatric STEC-HUS patients [360] and 44 adults [361]. Hemorrhagic complications are a rare event during the course of STEC-HUS, as highlighted by the fact that peritoneal and central venous catheter placement and removal can be accomplished safely in most cases without platelet transfusion [362]. Caution is advised, and indications should be restricted to active bleeding and invasive surgical procedures. In addition, allo-immunization may be an issue in patients with severe renal impairment who may need renal transplantation.

#### 6.2.5. Detrimental Effect of Antimotility Agents

Use of antimotility agents has been associated with an excess risk of HUS development in children infected with EHEC [236,247,363] and, in accordance with the 2014 recommendations of the European Society for Pediatric Gastroenterology for acute gastroenteritis [364], should therefore be discouraged. In spite of these recommendations, 21% of children and 43% of adults received antimotility agents during the course of the disease [344].

### 6.3. Specific Therapies

Over 30 years after the description of Shiga toxins and Shiga toxin *E. coli*-associated HUS [6], the quest for a specific treatment remains elusive, despite major achievements in the understanding of the pathophysiology of the disease.

#### 6.3.1. Plasma Exchange and Immunoadsorption

Soon after the original description of STEC-HUS, randomized trials tried to evaluate the indication of plasma exchange, based on the rationale that therapeutic plasma exchange could remove the toxin or circulating factors that damage the endothelium [365,366] and its efficacy in TTP [367]. These studies yielded contradictory results. The procedure was later evaluated in adults during the Lanarkshire outbreak [368] and in the O104:H4 epidemic, again yielding conflicting results. A small cohort study (n = 5) showed recovery of a normal neurological status in each patient at day 7 with early plasma exchange [369]. In another noncontrolled study performed on 12 severe cases, the initiation of immunoadsorption seemed to correlate with a favorable course of the neurological symptoms, prompting authors to envisage neurological complications in STEC-HUS as antibody-mediated [370]. However, a more methodologically sound case control study did not confirm the benefit of plasma exchange [329]. Pending randomized trials, the American Society for Apheresis and the Japanese Study Group for Hemolytic Uremic Syndrome both suggest the use of plasma therapy in a selected subgroup of STEC-HUS adult patients with severe neurological involvement [341,371] (weak and not graded recommendations, respectively).

#### 6.3.2. Complement Blockade Therapy

The first report of three STEC-HUS children treated with the humanized monoclonal antibody medication eculizumab was published in May 2011 and showed dramatic improvement in neurological status and discontinuation of dialysis after the first infusion [372]. Only a few days after this publication, the O104:H4 outbreak provided an unprecedented basis for clinical investigation, and many patients were treated with complement blockade therapy. If the analysis of the German registry [373] did not support the use of eculizumab in adult STEC-HUS cases, early treatment was associated with a rapid and efficient recovery in French patients [374] and children with central nervous system involvement [375]. Based on the conflicting results of these uncontrolled studies, a randomized multicenter controlled trial is currently under way (ECULISHU, NCT02205541).

#### 6.3.3. Gb3 Receptor Analogues, Shiga Toxin-Binding Agents, and Monoclonal Antibodies

Multiple efforts have been made to develop an agent that can bind and neutralize the Shiga toxin and thereby protect the endothelium. To date, this seemingly enticing prospect has failed to find its way in the modern armamentarium against STEC-HUS. Oral receptor analogues have been devised in a bid to bind the Shiga toxin before it translocates into the bloodstream and to entrap it in the gut. Several have been developed and tested in mice models [376,377,378], but so far only one has reached the clinical trial stage. SYNSORB Pk is an oral Shiga toxin binding agent that has been tested in an ambitious multicenter randomized trial including 150 children [273], which was prematurely stopped for futility after an interim analysis. The rapid development of endothelial insult following contact with Shiga toxins combined with its disappearance from patient stools and bloodstream by the time HUS develops [140,143] probably hinders the usefulness of such a strategy. Injectable Shiga toxin competitive inhibitors have also been developed in order to neutralize Shiga toxins once they cross the mucosal barrier. Named SUPER TWIG [379], STARFISH [380], PC7-30 [381], TF-1 [382], or DAISY [383], they have yet to translate into clinical applications, partly because of their synthetic complexity. Using a similar rationale, Paton et al. designed a recombinant bacterium expressing a Shiga toxin receptor mimic on its surface capable of adsorbing and neutralizing Shiga toxins, thus protecting mice from an otherwise lethal dose of STEC [384]. To date, this innovative approach has not been tested in humans, and its application to a piglet model failed to prove any clinical effect [385]. Monoclonal antibodies represent a powerful tool in modern biochemistry and drug development. Urtoxazumab is a humanized monoclonal antibody directed against Shiga toxin 2, which proved its efficacy in a gnotobiotic piglet model [386] and its tolerance in a phase 1 study in 2010 [387]. A randomized placebo-controlled study was conducted at the same time, but the results are still awaiting publication. Shigamabs^®^ comprises two chimeric monoclonal antibodies against Stx1 and Stx2. Its development was stopped after phase 1 [388] (phase 2 NCT01252199, unpublished). Several other monoclonal antibodies are under scrutiny, including single-chain antibodies [389,390], but they may face the same issue: a narrow time window for meaningful intervention because of the fleeting presence of circulating Shiga toxins.

#### 6.3.4. Manganese

Mukhopadhyay reported in 2012 that the widely available metal manganese was capable of blocking endosome-to-Golgi trafficking of Shiga toxins by targeting the cycling Golgi protein GPP130, thus causing its degradation in lysosomes [159]. Furthermore, mice perfused with manganese became completely resistant to a lethal Shiga toxin challenge. Even though the results have not been replicated by another team [391], and caution is advised regarding its potential neurological toxicity, manganese deserves further clinical evaluation owing to its low cost.

#### 6.3.5. Other Abandoned Therapies

Steroids [392] and antithrombotic therapies [393,394,395] have been abandoned due to the lack of proven benefit and the risk for secondary infection or bleeding, respectively.

## 7. Prognosis

Despite the relatively low lethality of pediatric STEC-HUS, which has fallen below 3% in recent years [105,218,346,396], EHEC remains a public health threat, as mortality can rise to 15%–33% in adult and fragile populations [107,108,215,397], and long-term sequelae have been increasingly described, affecting about one-third of patients [398].

### 7.1. Renal Sequelae

The kidneys bear the brunt of most of the long-term sequelae. According to a meta-analysis published in 2003 (including studies up to 2001), 12% of STEC-HUS patients included died or had end-stage renal disease, and 25% of patients had a glomerular filtration rate < 80 mL/min/1.73 m^2^, hypertension, or proteinuria after a mean 4.4 years of follow-up [399]. Prognosis seems to have improved, and a more recent study has recorded a lowered incidence of 1.4% of the patients receiving long-term RRT or transplantation. Nine percent had a glomerular filtration rate < 80 mL/min/1.73 m^2^ after a 5-year follow-up [240]. Compared with age-matched controls, this could represent an average decrease of 10 mL/min/1.73 m^2^ [400]. Mild renal abnormalities, namely proteinuria and hypertension, are described in 19% and 9% of the patients [240]. These are of particular concern, as they may contribute to chronic kidney disease and renal failure decades after the initial insult in otherwise healthy children, and they may have been underestimated in studies with only short and intermediate follow-up. Three years after the O104:H4 outbreak, chronic kidney disease, hypertension, and proteinuria were present in 4%, 19%, and 28% of the patients included in the German Pediatric HUS Registry, respectively [401]. In adults, chronic kidney disease, de novo hypertension, and proteinuria were observed in 47%, 25%, and 27% of the patients included in a single-center study [402]. The pathophysiology of chronic kidney disease after STEC-HUS is believed to be connected to the burden of hyperfiltration affecting the reduced pool of functional nephrons (rather than recurrence or persistence of the microangiopathic process) and progressive scarring [403]. This view is consistent with the development of focal segmental glomerulosclerosis and hyalinosis, which has been described [404]. This outcome is not specific to STEC-HUS, as the risk of chronic kidney disease following any given cause of AKI is increased in adult patients [405]. At any rate, these results stress the need for a long-term follow-up of STEC-HUS patients for a minimum of 5 years, including in cases of apparent full recovery [237,240]. For patients with renal sequelae after STEC-HUS, a low-sodium diet, early restriction of protein intake, and angiotensin-converting enzyme inhibitors in the event of hypertension or proteinuria seem to slow down the progression of chronic kidney disease [406,407]. If end-stage renal disease develops, renal transplantation in STEC-HUS patients appears to be safe, and the previously described recurrences after transplantation may have been linked to unrecognized complement mutations [408]. Two retrospective studies performed in Argentina, where STEC-HUS represent 15% of the causes leading to terminal renal failure in children who received transplantation [403], found excellent survival of patients and grafts, without evidence of recurrence in any patient [409,410]. As progression to end-stage renal disease is a rare event in the course of STEC-HUS, and given the risk of recurrence of atypical HUS [411], it seems reasonable to screen patients for complement mutations before renal transplantation.

### 7.2. Extra-Renal Sequelae

Next to renal sequelae, the most feared long-term complications after STEC-HUS are related to the central nervous system. Even if intellectual performance does not seem to be impaired in children [412], subtle neuromotor impairment has been described in most of them, independently of acute central nervous system involvement [413]. Neuropsychological symptoms, including fatigue, headache, and attention deficits, were present in 70% of adult patients 19 months after the O104:H4 outbreak [414]. Gastrointestinal complications, such as colonic stricture [415], are rare after the acute phase. Recurrence of diabetes warrants long-term screening of STEC-HUS survivors [258].

### 7.3. Predicting the Risk of Long-Term Sequelae

If 70% of EHEC-infected patients will fully recover, identification of the ones at risk for fatal outcome or long-term sequelae is of critical importance. Risk factors for death include young or old age [107], dehydration [243], elevated white blood cell count [243], and, perhaps most importantly, the presence of neurological symptoms [243,246]. Need and duration of dialysis are seemingly the most reliable predictors of poor renal outcome, and the occurrence of chronic sequelae increases stepwise with the duration of anuria [237,239,416,417,418]; an approximate 5% increase in the odds of renal sequelae with each supplementary day in dialysis has been reported [240]. Likewise, patients suffering from anuria for longer than 10 d are particularly vulnerable since normal renal function will not return in more than half of them. Higher leucocyte count and presence of hypertension may also be associated with poorer renal outcomes [112,240,418]. Extra-renal manifestations have sometimes been associated with a poorer long-term prognosis, probably as a reflection of the severity of the microangiopathic process [419]. Lastly, genetic factors also likely influence the course of the disease, and the 1166C allele of the angiotensin II type 1 receptor was found to have a significant protective effect [112]. Patients with normal renal function, normal blood pressure, and no proteinuria one year after the acute stage had an excellent prognosis, without occurrence of renal sequelae during further long-term follow-up [399].

## 8. Conclusions

In conclusion, Shiga toxin-associated HUS remains a global health concern. Our review emphasizes that data regarding adult patients are limited, and this scarcity probably prevents prompt recognition and implementation of the best standard of care for these patients. The emergence of new and more virulent pathogens such as the O104:H4 strain reminds us that, despite major improvements in the understanding of the pathophysiology and the encouraging results in preclinical models and ongoing clinical trials, a specific treatment is still absent, and concerns about sequelae make scientific and clinical research into this pathology a public health priority.

## Figures and Tables

**Figure 1 toxins-12-00067-f001:**
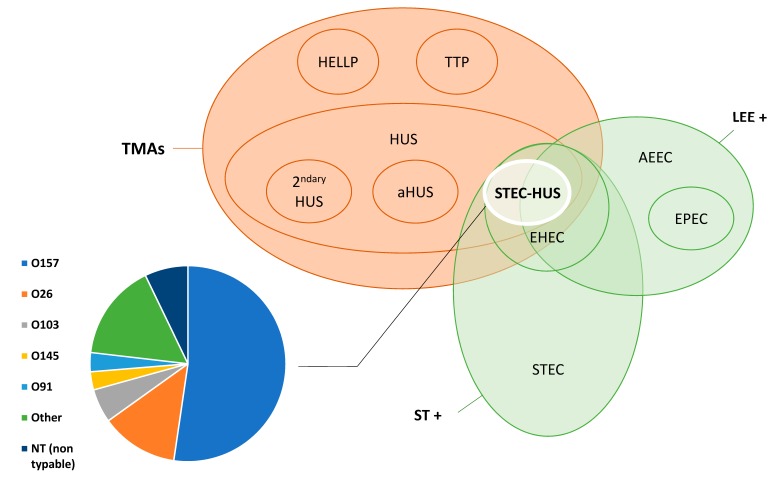
Nomenclature of thrombotic microangiopathies and pathogenic *Escherichia coli*, including distribution of serotypes in reported cases in 2012–2014 in Europe. Abbreviations—TMAs: thrombotic microangiopathies; HELLP: hemolysis, elevated liver enzymes and low platelets syndrome; TTP: thrombotic thrombocytopenic purpura; HUS: hemolytic uremic syndrome; aHUS: atypical hemolytic uremic syndrome; STEC-HUS: Shiga toxin *Escherichia coli*-associated hemolytic uremic syndrome; ST+: Shiga toxin-producing bacteria; EHEC: enterohemorrhagic *E. coli* (represent STEC serotypes pathogenic to humans); LEE+: locus of enterocyte effacement-expressing bacteria, *E. coli* expressing both ST and LEE genes (“typical STEC”); AEEC: attaching and effacing *E. coli*; EPEC: enteropathogenic *E. coli*. The distribution of EHEC serotypes corresponds to the reported cases in Europe between 2012 and 2014 [4].

**Figure 2 toxins-12-00067-f002:**
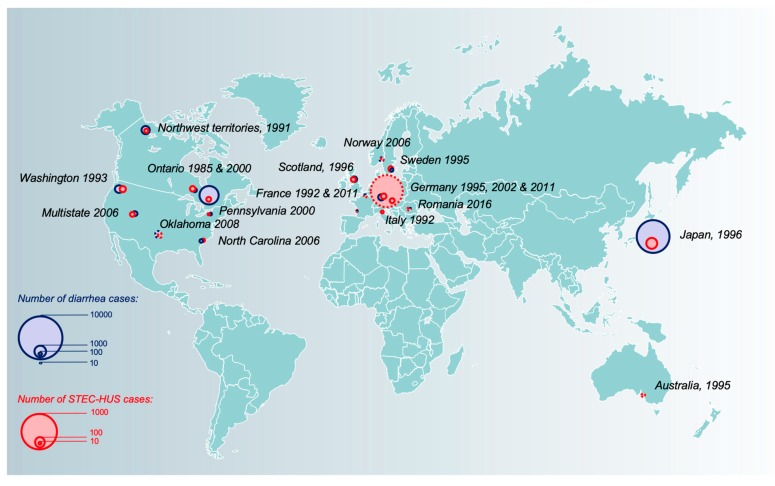
Proportional circles map of major outbreaks of enterohemorrhagic *E. coli* O157 and non-O157 reported in the literature (1985–2017). Published reports of outbreaks including more than 5 STEC-HUS cases are represented. The sizes of violet and red circles are proportional to the numbers of cases of diarrhea (when available) and hemolytic uremic syndrome reported in each outbreak, respectively, using perceptual scaling. Dashed circles represent outbreaks caused by non-O157 strains.

**Figure 3 toxins-12-00067-f003:**
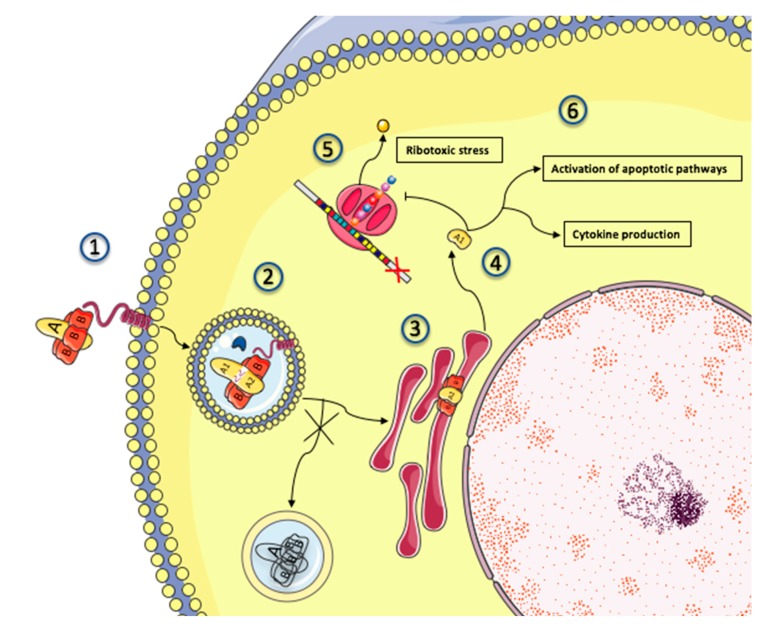
Intracellular trafficking and cytotoxicity of Shiga toxin. A simplified depiction of Shiga toxin intracellular trafficking and mechanisms of toxicity. 1: Shiga toxins consist of a monomeric enzymatically active A subunit, non-covalently linked to a pentameric B subunit. The B subunit binds to the glycosphingolipid globotriaosylceramide (Gb3), present in lipid rafts on the surface of the target cell. 2: Shiga toxin and its receptor are internalized (endocytosis), and Shiga toxin is activated through cleavage of the A subunit into 2 fragments by the protease furin (represented by a blue crescent). Disulfide bonds keep the 2 fragments together in the endosome. 3: Shiga toxin avoids the lysosomal pathway and is directed towards the endoplasmic reticulum (retrograde transport) where the disulfide bound is reduced. 4: The A1 subunit translocates into the cytoplasm (anterograde transport) where it can exert its cytotoxic effects. 5: The processed A1 fragment cleaves one adenine residue from the 28S RNA of the 60S ribosomal subunit, thus inhibiting protein synthesis and triggering the ribotoxic and endoplasmic reticulum stress responses. 6: In addition to its ribotoxic effect, Shiga toxin activates multiple stress signaling and apoptotic pathways, and is responsible for the production of inflammatory cytokines by target cells.

**Figure 4 toxins-12-00067-f004:**
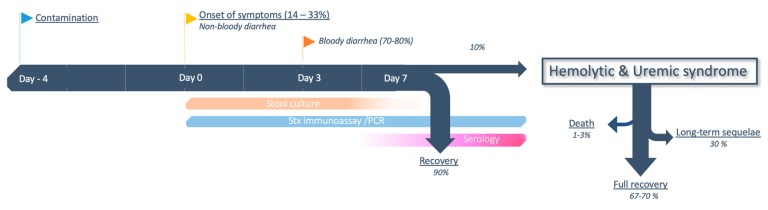
Timeframe of the development and evolution of STEC-HUS, with the theoretical window of diagnostic tests. The timeframe and proportions represented here are based on median values and are highly variable, depending on strain, epidemiological and individual patient characteristics.

**Table 1 toxins-12-00067-t001:** Common misconceptions about STEC-HUS.

**STEC-HUS mainly occurs through large outbreaks**
Despite sensational publications about large outbreaks, most STEC-HUS cases (≈75%) are actually sporadic, judging by nationwide studies [99] and surveillance networks [104].
**Ground beef is the cause of the majority of vehicle-born transmissions**
Cattle are a major reservoir for *E. coli*. Ground beef was responsible for the first outbreaks reported [6,7] and currently represents around 33% of cases [91].
***E. coli* is the only bacteria that produces Shiga toxin**
*Shigella dysenteriae* type 1 produces a chromosomally encoded toxin almost identical to Stx1 [217]. In addition, Stx phages can occasionally be found in other gram-negative bacteria (*Citrobacter*, *Salmonella*).
**Community-acquired nonbloody diarrhea does not suggest investigation for STEC**
If digestive symptoms are the rule in STEC infections, the proportion of bloody diarrhea can vary between 65%–80%, and is usually lower in non-O157 infections [41,218]. Investigations for STEC can be ordered for community-acquired diarrheas irrespective of the presence of blood [219].
**Complement is involved in the pathophysiology of atypical HUS, not STEC-HUS**
Although the breakthrough discovery of alternative complement pathway dysregulation in aHUS is not paralleled in STEC-HUS, recent publications highlighted a potential role in the pathophysiology of STEC-HUS [183], providing hope for potential clinical applications.
**HUS with a negative stool culture is probably atypical**
Stool culture sensitivity is insufficient to exclude STEC-HUS. The diagnostic strategy must include both culture and nonculture-based assays to detect Shiga toxins or the genes encoding it [219]. Additionally, by the time of HUS, enterohemorrhagic *E. coli* is less likely to be found in stool cultures [220].
**Antibiotics are detrimental during STEC infection**
Antibiotics are not recommended for STEC infection. Nevertheless, an important distinction has to be made between antibiotics capable of triggering bacterial SOS response and the release of Stx (fluoroquinolones, B-lactams) and others (azithromycin, fosfomycin) which do not [33,221]. The potential beneficial effects of the latter agents are currently being evaluated.
**O157:H7 is responsible for the majority of STEC infections throughout the world**
A shift in epidemiology occurred in the 2000s, and thanks to new diagnostic techniques, non-0157 serotypes are now more commonly found than 0157:H7 in Europe and North America [41,50]. However, 0157:H7 is still responsible for the majority of cases in Latin America [93].

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
