# Peer review of "Shiga Toxin-Associated Hemolytic Uremic Syndrome: A Narrative Review"

_toxins, 2020, doi:10.3390/toxins12020067_

Round 1

Reviewer 1 Report

Journal: Toxins

Manuscript ID: toxins-694409

Type of manuscript: Review

Title: SHIGA TOXIN-ASSOCIATED HEMOLYTIC UREMIC SYNDROME: A NARRATIVE REVIEW

The global emergence of virulent bacteria and especially E. coli related outbreaks and clinical diseases are a topic of great concern. Investigations on clinical pathogens and treatments are of great value to safeguard human health as well as to reduce socioeconomic burdens.

Here the authors present a review on Shiga-toxin associated HUS syndrome. The presented data are of interest to a wide readership with different backgrounds.

In general:

The manuscript is well formatted and written. The manuscript follows a logical order. I do not have any major concern. I do have a number of minor points.

Minor:

LN 5: additional factor = genetic plasticity and/or any extrachromosomal hereditary determinant

LN 50: 1983 was the first recorded outbreak – perhaps there had been outbreak not being diagnosed? Consider rewording.

LN 188: “… important …” is somewhat relative – perhaps “serious” - consider rewording.

LN 201: “during summer months” – there is some scientific evidence on STEC seasonality being more numerous during winter month. Perhaps ambient temperature and consumer behaviour (food handling) have a greater impact.

LN 272: “binds to”

LN 273: delete “very short”. It is less than 5 minutes, period.

LN 348: published results on those trials appear not to be so optimistic: https://www.ncbi.nlm.nih.gov/pubmed/31104069?dopt=Abstract

LN 380: SHU-STEC?

LN 386: Cattle are …

LN 407: Figure 4: inconclusive what the code/meaning of J-4 JO J3 J7 is – consider rewording.

LN 514: “Immunochemistry targeting the membrane attack complex …” consider rewording.

LN 571: Including my own diagnostic lab and scientific reports – with enrichments of 6 hours and subsequent PCR – data can be obtained in less than 12 hours.

LN 616: “ … such as monitoring the protease ADAMTS13 … “ consider rewording.

LN 646: consider rewording: cattle to ruminants – as others are carrier too.

LN 650 - Major: the authors should consider to add a paragraph 6.1.1.3 slaughter and meat processing as this is one of the key factors for food contaminations and food-borne outbreak, including preventions like hurdle technologies and such …

LN 720: “… with an odd of 2.38 …”

LN 730: very odd wording: “ … as well as imperceptible and digestive losses, …” – “… to monitor liquid and solid excreta …” perhaps?

LN 746: at first use – “Renal replacement therapy (RRT)

LN 749: “… is full blown, …” unprofessional wording?

LN 754: … no trial ever? How can you be certain – perhaps has been done but not reported …

LN 799: wording? Good outcome – was alive? Bad outcome – dead? Favorable (LN 809) outcome – half dead? Please consider a better description.

LN 806: meaning of “… (grade 2C …” likely remains unclear for readers

LN 809: consider rewording: “… following the administration of the humanized monoclonal antibody medication Eculizumab …"

Author Response

Reviewer 1

The global emergence of virulent bacteria and especially E. coli related outbreaks and clinical diseases are a topic of great concern. Investigations on clinical pathogens and treatments are of great value to safeguard human health as well as to reduce socioeconomic burdens.

Here the authors present a review on Shiga-toxin associated HUS syndrome. The presented data are of interest to a wide readership with different backgrounds.

In general:

The manuscript is well formatted and written. The manuscript follows a logical order. I do not have any major concern. I do have a number of minor points.

We would like to thank reviewer 1 for these kind comments.

We hope the quality of the revised version of our manuscript has been enhanced based on his/her remarks and suggestions.

Here are a point by point response :

Minor:

LN 5: additional factor = genetic plasticity and/or any extrachromosomal hereditary determinant

LN 50: 1983 was the first recorded outbreak – perhaps there had been outbreak not being diagnosed? Consider rewording.

Indeed it is more than probable that previous outbreaks had not been reported. We added the word « recorded [outbreaks] » to make it clear to the reader.

LN 188: “… important …” is somewhat relative – perhaps “serious” - consider rewording.

Thank you, the modification has been done.

LN 201: “during summer months” – there is some scientific evidence on STEC seasonality being more numerous during winter month. Perhaps ambient temperature and consumer behaviour (food handling) have a greater impact.

The articles we refer to (Elliott, Nationwide study of haemolytic uraemic syndrome: clinical, microbiological, and epidemiological features. Arch Dis Child. 2001 & Griffin, The Epidemiology of Infections Caused by Escherichia coli O157: H7, Other Enterohemorrhagic E. coli , and the Associated Hemolytic Uremic Syndrome. Epidemiol Rev. 1991) describe a higher incidence of STEC in summer months. This may indeed be due to consumer behaviour and ambient temperature. If you have opposite evidence from the literature we can temper or delete this assertion.

LN 272: “binds to”

Thank you, this error has been corrected

LN 273: delete “very short”. It is less than 5 minutes, period.

Indeed this somehow introduced subjectivity into the sentence and was deleted.

LN 348: published results on those trials appear not to be so optimistic: https://www.ncbi.nlm.nih.gov/pubmed/31104069?dopt=Abstract

Thank you for this information, the results have been published while we were writing this review. We included the results and the reference in our manuscript. In our view, the fact that ART-123 did not reduce all cause mortality in disseminated intravascular coagulation is not indicative of its potential as a future treatment for STEC-HUS and we have not changed our last sentence about its first encouraging results in this setting.

LN 380: SHU-STEC?

Thank you, this mistake has been corrected.

LN 386: Cattle are …

Thank you, this mistake has been corrected.

LN 407: Figure 4: inconclusive what the code/meaning of J-4 JO J3 J7 is – consider rewording.

We changed this code to Day-4, Day 0, Day 3 and Day 7. Sorry for this mistake.

LN 514: “Immunochemistry targeting the membrane attack complex …” consider rewording.

The article we are referring to mentions immunohistochemical staining for both the membrane attack complex and other fractions of complement (C1q, C3, C4d). If you agree, we would prefer to keep this sentence as it is currently written.

LN 571: Including my own diagnostic lab and scientific reports – with enrichments of 6 hours and subsequent PCR – data can be obtained in less than 12 hours.

Thank you. This has been corrected.

LN 616: “ … such as monitoring the protease ADAMTS13 … “ consider rewording.

This sentence has been reworded according to your suggestion.

LN 646: consider rewording: cattle to ruminants – as others are carrier too.

Thank you for this precision. We corrected our sentence accordingly.

LN 650 - Major: the authors should consider to add a paragraph 6.1.1.3 slaughter and meat processing as this is one of the key factors for food contaminations and food-borne outbreak, including preventions like hurdle technologies and such …

We agree slaughter hygiene and meat processing importance was underrepresented in the manuscript. We added a short paragraph highlighting its importance

LN 720: “… with an odd of 2.38 …”

Thank you for pointing this out. We corrected this sentence.

LN 730: very odd wording: “ … as well as imperceptible and digestive losses, …” – “… to monitor liquid and solid excreta …” perhaps?

We simplified this sentence.

LN 746: at first use – “Renal replacement therapy (RRT)

Thank you for this precision, we corrected this.

LN 749: “… is full blown, …” unprofessional wording?

We replaced this expression by “established”.

LN 754: … no trial ever? How can you be certain – perhaps has been done but not reported …

After extensive search through Pubmed but also ClinicalTrials.gov and the ISRCTN registry, we could find no trace of such study. This is understandable due to the difficulties of this kind of study and apart from volume expansion, prospective studies on STEC-HUS all concerned investigational drugs.

LN 799: wording? Good outcome – was alive? Bad outcome – dead? Favorable (LN 809) outcome – half dead? Please consider a better description.

We agree the wording was not precise enough. Thank you for pointing it out. We now report the exact outcomes from the references cited.

LN 806: meaning of “… (grade 2C …” likely remains unclear for readers

We changed grade 2C to “weak”, as described by the GRADE group.

LN 809: consider rewording: “… following the administration of the humanized monoclonal antibody medication Eculizumab …"

Thank you for the suggestion. We added the information about the nature of Eculizumab in the first sentence of paragraph 6.3.2 : “The first report of three STEC-HUS children treated with humanized monoclonal antibody medication Eculizumab was published in May 2011…”

Reviewer 2 Report

This manuscript is an extensive review on all aspects of STEC and STEC associated infectious diseases. Although this is very well written and there is almost no criticism in the contents, following minor revisions are necessary.

line 51: "Shiga toxin E. coli" should read "Shiga toxin-producing E. coli". line 56, 57: E. coli should be italicized. line 58: In the word of Stx phage, "Stx" appears first time. When any abbreviation appears first time, full spelling of the word should be also written, as authors described elsewhere. line 68: AggR characteristics should be briefly shown or its full spelling should be written. line 100: should read "higher case-fatality (11%)". Fig.2: Explanations of marks in this Figure, shown in the lower left corner are too small to read. Expand this portion. line 130: should read "the sizes". line 130-132: ", respectively" should be added. line 184-185: Is it the case for non-STEC? Otherwise, STEC can survive for months in environment, while common E. coli cannot? line 222: keep space in "1balpha". line 239-241: meaning of the sentence is not clear. Rephrase so that readers can easily understand the meaning. line 259: consider rephrase as "if it was not due to Shiga toxin". 273-274: Why half-life of Stx is so short? Because of degradation in serum, or clearance in liver? Any reason should be added briefly. Fig.3: In this figure, there is a small, blue, crescent-like mark. Dies it mean furin? If so, it is better to add this name in the Figure, or any indication in figure legend. 

Author Response

Reviewer 2

This manuscript is an extensive review on all aspects of STEC and STEC associated infectious diseases. Although this is very well written and there is almost no criticism in the contents, following minor revisions are necessary.

line 51: "Shiga toxin E. coli" should read "Shiga toxin-producing E. coli".

This has been corrected.

line 56, 57: E. coli should be italicized. line 58: In the word of Stx phage, "Stx" appears first time. When any abbreviation appears first time, full spelling of the word should be also written, as authors described elsewhere.

Thank you for pointing out this mistake. It has been corrected.

line 68: AggR characteristics should be briefly shown or its full spelling should be written.

Thank you. We precised the role of AggR as a transcriptional regulator. If you agree we do not wish to go into further details to remain concise about microbiological traits that are not specific to EHEC.

line 100: should read "higher case-fatality (11%)".

Thank you for this correction.

Fig.2: Explanations of marks in this Figure, shown in the lower left corner are too small to read. Expand this portion.

We agree the legend was not easily readable. We increased the size of the text to make it easier. The size of the circles, however, is representative of the circles displayed upon the map and cannot be changed without significant overlapping of the outbreaks.

line 130: should read "the sizes".

Thank you for pointing out this mistake. We corrected it.

line 130-132: ", respectively" should be added.

Thank you. We added it.

line 184-185: Is it the case for non-STEC? Otherwise, STEC can survive for months in environment, while common E. coli cannot?

Persistence in the environement is the case for most E. coli. Thank you for this precision.

line 222: keep space in "1balpha".

This has been corrected.

line 239-241: meaning of the sentence is not clear. Rephrase so that readers can easily understand the meaning.

We agree this sentence was not very clear and a bit technical. We believe the difficulties of creating a relevant model for STEC-HUS is an important point to highlight for the reader, and we hope that we rephrased our sentence in a way that make its reading easier.

line 259: consider rephrase as "if it was not due to Shiga toxin".

Thanks. We rephrased the sentence according to your suggestion.

lines 273-274: Why half-life of Stx is so short? Because of degradation in serum, or clearance in liver? Any reason should be added briefly.

Half-life of Stx is very short in the serum, but significantly longer in tissues. We made this precision which we agree was needed in our sentence. Thank you for your question and suggestion.

Fig.3: In this figure, there is a small, blue, crescent-like mark. Dies it mean furin? If so, it is better to add this name in the Figure, or any indication in figure legend.  

Yes, the crescent-like mark represents furin. For reasons of space, it is difficult to add annotation in the figure, but we specified it in the legend. Thanks again for your suggestion.

Reviewer 3 Report

Journal: Toxins

Manuscript title: Shiga toxin-associated hemolytic uremic syndrome: a narrative review

Manuscript ID: toxins-694409

In this manuscript, a review on the current knowledge of Shiga toxin-producing Escherichia coli-associated hemolytic uremic syndrome (STEC-HUS) was carried out. STEC-HUS belongs to the body of thrombotic microangiopathies (1), a heterogeneous group of diseases characterized by a triad of features: thrombocytopenia, mechanical hemolytic anemia with schistocytosis and ischemic organ damage. It is caused by gastrointestinal infection by a Shiga toxin-producing E. coli (and occasionally other pathogens) and is also called “typical” HUS, as opposed to “atypical” HUS which results from alternative complement pathway dysregulation, and “secondary” HUS, caused by various co-existing conditions.

The subject of the review iso f great interest and can actract interest in a medical context, in public health area, and from the social point of view.

The subject is treated with particolar attention to the adult age, whereas, in general, more information are available on infancy effects of these diseases.

The review is very interesting, treating epidemiology and microbiology, STEC-HUS as a zoonosis, pathogenesis, diagnosis, treatment, prognosis. It is well written and it is informative for persons working in different sectors, always related to health. The review traits the Shiga toxin-associated HUS as a global health concern.

The novelty of the review deals with diffusion of aspects related to STEC-HUS in adult patients, that are limited, preventing implementation of care for these patients. Moreover, an important aspect of interest of the review is the analysis of the the emergence of new and more virulent pathogens such as the O104:H4 strain, highlighting important aspects of public health.

In the Abstract section, the fact that the main interest of this review focuses on Shiga toxin-producing Escherichia coli-associated hemolytic uremic syndrome (STEC-HUS) is less emphasized. Actually, the “Key contribution” section reports clearly the aims of the review, probably a little mention in the Abstract could be informative. The Title of the review underlines the STEC-HUS aspects, in particolar in adults, whereas in the Abstract section, STEC is the main focus, without mentioning STEC-HUS aspects. In the abstract, specify and highlight the main aspects of the review.

Revisions

Line 14: “Escherichia coli” change to Italic style and throughout the manuscript;

Line 33: “… Vero cells (a strain of renal epithelial cells …)” change to ‘… Vero cells (a line of renal epithelial cells …)’;

Line 57: “Escherichia coli” change to Italic style and throughout the manuscript;

Line 58: “… by a Stx phage …” change to ‘… by means of a Stx phage …’;

Line 79: “S. dysenteriae type 1” change ‘type 1’ not in Italic style;

Line 346: “… (194),” change to ‘“… (194)’;

Line 386: In Table 1 References are reported by names of Authors (Elliot Arch Dis Child 2001), whereas the Reference list include numbers (99. Elliott EJ, Robins-Browne RM, O’Loughlin EV, Bennett-Wood V, Bourke J, Henning P, et al. Nationwide study of haemolytic uraemic syndrome: clinical, microbiological, and epidemiological features. Arch Dis Child. 2001 Aug;85(2):125–31.). Please, uniform.

Lines 602 and 603: “… the possibility of cross-reactions with other bacterial strains (Salmonella, Yersinia, Citrobacter),” change to ‘… the possibility of cross-reactions with other bacterial strains belonging to different genera (Salmonella, Yersinia, Citrobacter),;

Line 936: Please, include year in reference number 3;

Lines 1296 and 1297: Abbreviate first name of Authors;

In the Reference section change names of bacterial species in Italic style and uniform capital letters in the title of references, thoughout the manuscript.

Manuscript title: Shiga toxin-associated hemolytic uremic syndrome: a narrative review

Manuscript ID: toxins-694409

In this manuscript, a review on the current knowledge of Shiga toxin-producing Escherichia coli-associated hemolytic uremic syndrome (STEC-HUS) was carried out. STEC-HUS belongs to the body of thrombotic microangiopathies (1), a heterogeneous group of diseases characterized by a triad of features: thrombocytopenia, mechanical hemolytic anemia with schistocytosis and ischemic organ damage. It is caused by gastrointestinal infection by a Shiga toxin-producing E. coli (and occasionally other pathogens) and is also called “typical” HUS, as opposed to “atypical” HUS which results from alternative complement pathway dysregulation, and “secondary” HUS, caused by various co-existing conditions.

The subject of the review iso f great interest and can actract interest in a medical context, in public health area, and from the social point of view.

The subject is treated with particolar attention to the adult age, whereas, in general, more information are available on infancy effects of these diseases.

The review is very interesting treating epidemiology and microbiology, STEC-HUS as a zoonosis, pathogenesis, diagnosis, treatment, prognosis. It is well written and it is informative for persons working in different sectors, always related to health. The review traits the Shiga toxin-associated HUS as a global health concern.

The novelty of the review deals with diffusion of aspects related to STEC-HUS in adult patients, that are limited, preventing implementation of care for these patients. Moreover, an important aspect of interest of the review is the analysis of the the emergence of new and more virulent pathogens such as the O104:H4 strain, highlighting important aspects of public health.

In the Abstract section, the fact that the main interest of this review focuses on Shiga toxin-producing Escherichia coli-associated hemolytic uremic syndrome (STEC-HUS) is less emphasized. Actually, the “Key contribution” section reports clearly the aims of the review, probably a little mention in the Abstract could be informative. The Title of the review underlines the STEC-HUS aspects, in particolar in adults, whereas in the Abstract section, STEC is the main focus, without mentioning STEC-HUS aspects. In the abstract, specify and highlight the main aspects of the review.

Revisions

Line 14: “Escherichia coli” change to Italic style and throughout the manuscript;

Line 33: “… Vero cells (a strain of renal epithelial cells …)” change to ‘… Vero cells (a line of renal epithelial cells …)’;

Line 57: “Escherichia coli” change to Italic style and throughout the manuscript;

Line 58: “… by a Stx phage …” change to ‘… by means of a Stx phage …’;

Line 79: “S. dysenteriae type 1” change ‘type 1’ not in Italic style;

Line 346: “… (194),” change to ‘“… (194)’;

Line 386: In Table 1 References are reported by names of Authors (Elliot Arch Dis Child 2001), whereas the Reference list include numbers (99. Elliott EJ, Robins-Browne RM, O’Loughlin EV, Bennett-Wood V, Bourke J, Henning P, et al. Nationwide study of haemolytic uraemic syndrome: clinical, microbiological, and epidemiological features. Arch Dis Child. 2001 Aug;85(2):125–31.). Please, uniform.

Lines 602 and 603: “… the possibility of cross-reactions with other bacterial strains (Salmonella, Yersinia, Citrobacter),” change to ‘… the possibility of cross-reactions with other bacterial strains belonging to different genera (Salmonella, Yersinia, Citrobacter),;

Line 936: Please, include year in reference number 3;

Lines 1296 and 1297: Abbreviate first name of Authors;

In the Reference section change names of bacterial species in Italic style and uniform capital letters in the title of references, thoughout the manuscript.
